# Unveiling Markov Heads in Pretrained Language Models for Offline Reinforcement Learning

**Wenhao Zhao**[1][*][†]  **Qiushui Xu**[2][*][†]  **Linjie Xu**[3][†]  **Lei Song**[4]  **Jinyu Wang**[4]  **Chunlai Zhou**[1]  **Jiang Bian**[4]

## Abstract

Recently, incorporating knowledge from pretrained language models (PLMs) into decision transformers (DTs) has generated significant attention in offline reinforcement learning (RL). These PLMs perform well in RL tasks, raising an intriguing question: what kind of knowledge from PLMs has been transferred to RL to achieve such good results? This work first dives into this problem by analyzing each head quantitatively and points out *Markov head*, a crucial component that exists in the attention heads of PLMs. It leads to extreme attention on the last-input token and performs well only in short-term environments. Furthermore, we prove that this extreme attention cannot be changed by re-training embedding layer or fine-tuning. Inspired by our analysis, we propose a general method `GPT2-DTMA`, which equips a pretrained DT with Mixture of Attention (MoA), to accommodate diverse attention requirements during fine-tuning. Extensive experiments corroborate our theorems and demonstrate the effectiveness of `GPT2-DTMA`: it achieves comparable performance in short-term environments while significantly narrowing the performance gap in long-term environments.

## 1. Introduction

Transformers (Vaswani et al., 2017) achieve significant improvements in natural language processing (Devlin et al., 2018), computer vision (Yuan et al., 2021) and AI4Science (Wang et al., 2024) tasks, for it encodes the input data into powerful features via the attention mechanism. Applying transformers to the field of reinforcement learning (RL), Decision Transformer (DT) (Chen et al., 2021) which models the offline RL problem (Levine et al., 2020) to a return-conditional sequence-to-sequence problem, has demonstrated superior performance in solving different types of RL problems (Correia & Alexandre, 2023; Mezghani et al., 2023; Badrinath et al., 2024; Janner et al., 2021).

Recently, using pretrained language models (PLMs) and fine-tuning them for specific tasks (Kenton & Toutanova, 2019; Raffel et al., 2020) has gradually replaced training Transformers from scratch. Pretraining phase enables PLMs to gain rich universal language representations transferable to various downstream tasks (Xiong et al., 2024). Remarkably, Noorbakhsh et al. (2021); Goel et al. (2022) showed that PLMs can effectively tackle tasks beyond NLP. Inspired by the powerful knowledge transfer capabilities of PLMs, Reid et al. (2022); Shi et al. (2024) attempt to transfer NLP knowledge to RL domain, indicating that PLMs can enhance the performance of downstream offline RL tasks as well. We collectively refer to the DT variant models, which are initialized with PLMs, as `GPT2-DT`.

To understand this surprising efficacy when transferring knowledge from PLMs to offline RL, Shi et al. (2024) concluded that sequential modeling ability is the key to unleashing the potential of PLMs for RL tasks. Furthermore, Takagi (2022) proposes a hypothesis that pretraining with text is likely to make PLMs get context-like information and utilize it to solve the downstream task. However, it is still unclear that why these key features are essential for RL tasks and how to acquire these knowledge through pretraining. To bring clarity to PLMs in offline RL and provide more insightful guidance on how to better train and adapt transformer-based models for RL tasks, we dive into the mechanism of PLMs and do more quantitative analysis on the direct transferred parameters to RL. Based on our quantitative analysis, we first propose the definition of *Markov heads*, which inherit from PLMs and are fundamental for RL tasks and then adapt PLMs to all kinds of RL tasks by flexibly utilizing the properties of *Markov heads*.

---

[*]Equal contribution; the order is decided by a dice roll.[†]This work was conducted when the author was an intern at Microsoft Research Asia. [1]Computer Science Department, Renmin University of China, Beijing, China [2]Department of Industrial and Manufacturing Engineering, Pennsylvania State University, University Park, PA,USA [3]School of Electrical Engineering and Computer Scientce, Queen Mary University of London, London, UK [4]Microsoft Research Asia, Beijing, China. Correspondence to: Lei Song <lei.song@microsoft.com>.

*Proceedings of the 42$^{nd}$ International Conference on Machine Learning*, Vancouver, Canada. PMLR 267, 2025. Copyright 2025 by the author(s).

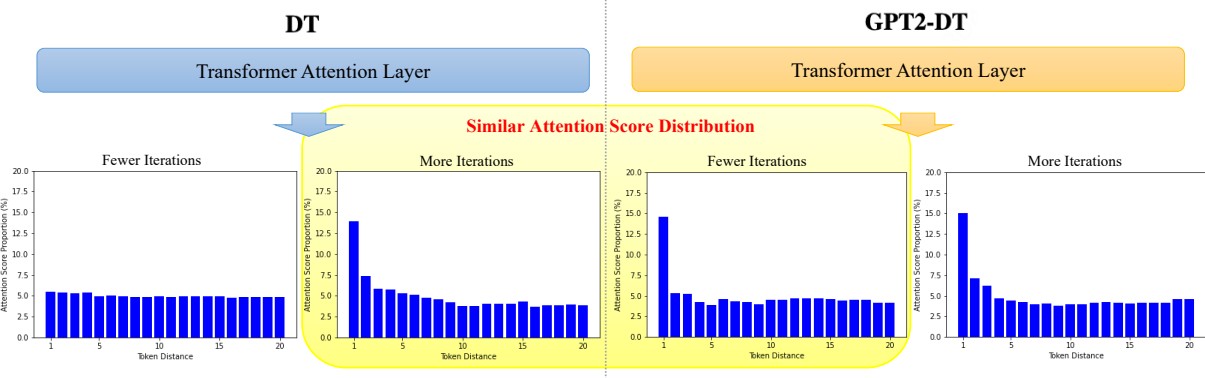

*Figure 1.* We compare the attention score distribution of `DT` and `GPT2-DT` during the process of fine-tuning within fewer iterations and more iterations under Hopper-medium environment. More comparisons of the attention score distribution on Walker2d-medium and Halfcheetah-medium can be found in Appendix B.

In this work, we investigate both short-term and long-term planning ability (Shang et al., 2022) of PLMs. To be more specific, we define *short-term* environments as where the actions prediction recalls a small amount of contextual information from most recent timesteps (e.g. MuJoCo Locomotion), and *long-term* environments that require the agent to extract useful information from the whole previous context (e.g. PointMaze), as context from far-off timesteps is helpful to predict reasonable actions under this scenario. The reason why we want to consider both short-term and long-term is that, our investigation found while PLMs perform better in short-term environments, it is less effective in long-term environments compared to a DT trained from scratch. Therefore, we are interested in what kind of NLP knowledge from PLMs has been successfully transferred to short-term RL tasks but failed in long-term RL task.

We discover that, as shown in Figure 1, `GPT2-DT` (Reid et al., 2022; Shi et al., 2024; Yang & Xu, 2024) has shared similar attention score distribution with well-trained `DT`, even at the initial stage of its training. We define the importance score of each head by zero-out each head and measuring the difference of the predictions. We also demonstrated the heatmap of each head and found that all attention heads can be divided into two categories. In particular, we define *Markov heads* as one kind of special heads inheriting from NLP tasks and identify that these heads lead DTs to focus more on the most recent state in short-term RL tasks by our theoretical analysis. Furthermore, we examine that *Markov heads* will be retained under any random embedding initializations and cannot be easily switched to *Non-Markov heads* by fine-tuning both theoretically and practically.

Naturally, we are also curious if we can still adapt PLMs to all kinds of RL tasks without training from scratch even though *Markov heads* will not be changed by fine-tuning. To bridge the performance gap in *long-term* environments, we propose `GPT2-DTMA`, combining PLMs with Mixture of Attention (MoA). Here, we consider each attention head as an expert and learn the weights of each expert adaptively according to each downstream RL task, so that `GPT2-DTMA` can automatically identify the required planning abilities by adjusting the weights of *Markov heads* when predicting the next action. `GPT2-DTMA` achieves comparable performance in short-term environments and narrows performance gap in long-term environments.

In summary, our contributions are as follows:

- We first propose the definition of *Markov heads* and further explore the properties of *Markov heads*. We prove that *Markov heads* will be preserved under any embedding initializations and fail to modify by fine-tuning. We also validate our theorems through extensive comparative experiments.

- We do a comprehensive quantitative analysis on the heads of PLMs by using different ways to explore the deeper mechanism of PLMs based on our definitions and theoretical analysis. We show that *Markov heads* are the key information transferred from NLP.

- Finally, we propose a general model `GPT2-DTMA` that can learn the weights of each head automatically and it significantly narrowed the performance gap of our model between short-term and long-term RL tasks compared to baselines.

## 2. Related Work

### 2.1. Decision Transformer

Transformer-based architecture (Vaswani et al., 2017) is widely explored in offline RL. Decision Transformer (DT) (Chen et al., 2021) firstly models RL as an autoregressive

generation problem. By conditioning on previous trajectories and returns-to-go, DT can generate desired actions without explicit reward modeling or dynamic programming. Furthermore, Furuta et al. (2021) explores other kinds of hindsight information instead of returns-to-go that can benefit sequential decision-making. Q-learning DT (Yamagata et al., 2023) proposes to combine DT with dynamic programming by utilizing a conservative value function to relabel returns-to-go in the training dataset. In addition to modifying the context by augmented information (Wang et al., 2024), some other work is also attempting to modify the model architecture of DTs. Shang et al. (2022) argues that the DT structure, which requires all tokens as input, is inefficient for learning Markovian-like dependencies. Kim et al. (2023) proposes replacing the attention layers with convolutional layers to better capture the inherent local dependent patterns in trajectories of RL tasks. However, all existing work focuses mainly on data augmentation for whole offline datasets and model redesign. These methods are still struggling with low computation-efficiency and unstable performance on all RL tasks. In our work, we aim at without modifying the model architecture or introducing new hidden information, and prove that PLMs can effectively capture local information thanks to *Markov heads*.

## 2.2. Pretrained Methods for Offline RL

A variety of approaches have been proposed to leverage pretraining technique in offline reinforcement learning. Nair et al. (2020); Zheng et al. (2022); Kostrikov et al. (2021) follow the standard pretrain-finetune paradigm, pretraining on a single-task offline dataset followed by fine-tuning in an online environment. Xie et al. (2023) pretrain decision transformers (DTs) using reward-free trajectories, and then fine-tune the model using reward-tagged trajectories collected online. Additionally, pretraining on mixed data collected from various RL tasks (Liu et al., 2022; Xu et al., 2022; Sun et al., 2022; Carroll et al., 2022; Wu et al., 2023) can achieve good performance on downstream tasks through fine-tuning or few-shot learning.

However, the scale of pretraining datasets with trajectory structures in RL domain is not comparable to the scale of text-structured datasets in NLP domain, which limits DTs' ability to thoroughly explore the nature of RL tasks. To address this, a pretraining paradigm on language datasets has been proposed; more details can be found in Appendix A. Reid et al. (2022); Shi et al. (2024) suggest that initializing DTs with pretrained parameters from PLMs, such as those from GPT-2, can overcome this limitation and outperform DTs that are initialized randomly. Yang & Xu (2024); Zheng et al. (2024) also utilize text content as pretraining samples, but during the fine-tuning phase, they incorporate the concept of Prompt-DT (Xu et al., 2022) by using trajectories from the single-task or multi-tasks as prior input

information to assist DTs in action prediction.

## 3. Preliminaries

### 3.1. Offline Reinforcement Learning

In offline RL, the policy model $\pi$ is trained on a pre-collected dataset, rather than interacting with the environment in real time. We consider learning in a Markov decision process (MDP) described by $\mathcal{M} = \langle \rho_0, \mathcal{S}, \mathcal{A}, P, \mathcal{R}, \gamma \rangle$, where $\rho_0$ is the initial state distribution, $\mathcal{S}$ is the state space, $\mathcal{A}$ is the action space, $P(s'|s, a)$ is the transition probability, $\mathcal{R}(s, a)$ is the reward function and $\gamma \in (0, 1]$ is the discount factor. We use $s_t, a_t$ and $r_t = \mathcal{R}(s_t, a_t)$ to denote the state, action and reward at timestep $t$, respectively. The goal of offline RL is to find an optimal policy $\pi^*$ that maximizes the $\gamma$-discounted expected return:

$$\max_{\pi} E_{s_{0:T}, a_{0:T} \sim \rho_0, \pi, P}\Big[\sum_{t=0}^{T} \gamma^t R(s_t, a_t)\Big]. \quad (1)$$

### 3.2. Decision Transformer

DTs (Chen et al., 2021) are designed to generate next actions relied on the information about expected future rewards for the current trajectories, therefore they propose returns-to-go (RTG) denoted as $\hat{R}_t \triangleq \sum_{i=t}^{T} r_i$, which represents the sum of expected future rewards from the current timestep $t$. In order to incorporate RL problems into a sequential model, DTs transform the trajectory into the format of $\tau = (\hat{R}_0, s_0, a_0, \hat{R}_1, s_1, a_1, \ldots, \hat{R}_T, s_T, a_T)$, where $\hat{R}_t, s_t, a_t$ are the RTG, state and action at timestep $t$, respectively. The next action predicted by policy $\pi_\theta$ is conditioned on the previous trajectory $\tau$ up to the timestep $t - 1$, current RTG $\hat{R}_t$ and current state $s_t$:

$$a'_t = \pi_\theta(\hat{R}_0, s_0, a_0, \ldots a_{t-1}, \hat{R}_t, s_t), \quad (2)$$

and $\pi_\theta$ is trained by minimizing the squared error between the true action $a_t$ and the predicted next action $a'_t$:

$$\mathcal{L}_{DT} = \sum_{t=0}^{T} \|a_t - a'_t\|_2^2. \quad (3)$$

## 4. Methodology

In this section, we will first give the definition of *Markov heads* and analyze the properties of *Markov heads*. Based on our theoretical analysis, we then introduce a general approach `GPT2-DTMA`, that shows great adaptive abilities to either long-term or short-term environments.

### 4.1. Markov Head

**Definition 4.1.** For any matrix $A \in \mathbb{R}^{d \times d}$, $A$ is a *Markov matrix* if and only if $A$ satisfies the following conditions:

(i). The diagonal elements $\{A_{i,i}\}_{i=1}^d$ are all positive.

(ii). The ratio between the mean of absolute diagonal elements and the mean of absolute off-diagonal elements are strictly greater than some large positive number $r$, i.e., $\frac{\overline{|A_{ii}|}}{\overline{|A_{ij}|}} > r$, where $\overline{|A_{ii}|} = \frac{\sum_{i=1}^d |A_{ii}|}{d}$ and $\overline{|A_{ij}|} = \frac{\sum_{i,j=1,\cdots,d,i\neq j} |A_{ij}|}{d^2-d}$.

For each head, we have three weight matrices corresponding to key, query and value. According to the definition of attention scores, we will focus our analysis on $W^q(W^k)^T$, which is the matrix multiplication between key weight matrix and query weight matrix. Then, we introduce the definition of *Markov head* based on Definition 4.1.

**Definition 4.2.** For head $i$, if the weight matrix $W_i^q(W_i^k)^T$ is a *Markov matrix*, then we say head $i$ is a *Markov head*.

### 4.1.1. THEORETICAL ANALYSIS

*Markov matrix* has some important properties that can give us more insight about what knowledge in PLMs is beneficial for RL tasks or limit the performance of generalization ability in RL tasks. In this section, we provide a theoretical analysis and a detailed explanation of the properties of *Markov matrix* and *Markov heads*.

**Theorem 4.3.** *For any random embedding vector $\mathbf{e}_i \in \mathbb{R}^{1\times d}, i = 1, \cdots, K$, the elements of each $\mathbf{e}_i$ are i.i.d. sampled from a normal distribution. For any given Markov matrix $A \in \mathbb{R}^{d\times d}$ satisfying Definition 4.1, then $\mathbb{E}[EAE^T]$ is also a Markov matrix, where $E = (\mathbf{e}_1, \cdots, \mathbf{e}_K)^T \in \mathbb{R}^{K\times d}$.*

*Proof.* See more details in Appendix E.1. $\square$

Based on Theorem 4.3, we extend the expectation result to a high probability bound. As a corollary, we show that for any random embedding matrix $E$, any given Markov matrix $A$ satisfies Definition 4.1 with high probability.

**Corollary 4.4.** *Define the diagonal mean and off-diagonal mean of $\Pi \triangleq EAE^T$ as*

$$D \triangleq \frac{1}{K}\sum_{i=1}^K |\Pi_{ii}|, \quad O \triangleq \frac{1}{K(K-1)}\sum_{i\neq j} |\Pi_{ij}|. \quad (4)$$

*Given a Markov matrix $A \in \mathbb{R}^{d\times d}$ satisfying Definition 4.1, for any $\varepsilon \triangleq \frac{\mathbb{E}[D]-r\mathbb{E}[O]}{2r} > 0$, there exists constants $c_1, c_2 > 0$ such that*

$$\mathbb{P}\left(\frac{D}{O} > r\right) \geq 1 - 2e^{(-c_1 K\tilde{\epsilon})} - 2e^{(-c_2 K(K-1)\tilde{\epsilon})}, \quad (5)$$

*where $\tilde{\epsilon}$ is defined as $\min(\frac{\varepsilon^2}{\|A\|_F^2}, \frac{\varepsilon}{\|A\|})$, and $\|A\|_F$ is the Frobenius norm of Markov matrix $A$, defined as $\|A\|_F \triangleq \sqrt{\sum_{i=1}^m\sum_{j=1}^n |A_{ij}|^2}$.*

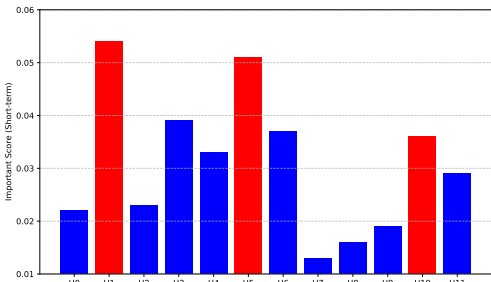

*Figure 2.* Importance score of each head of `GPT2-DT` in Hopper-medium environment. The red bar represents the importance scores allocated to the *Markov head* in the first attention layer. We also demonstrate importance score in PointMaze-large in Appendix D.

*Proof.* See more details in Appendix E.2. $\square$

In the next theorem, we can show that *Markov matrix* will be maintained after finite-time fine-tuning.

**Theorem 4.5.** *Suppose all the norm of the gradients are upper bounded by some positive number $B$ and the learning rate $\eta_k$ has a uniformly upper bound $\eta_0$. For any Markov matrix $A^0$, for any training iterations $K < \lfloor\min\left\{\frac{\rho}{r+1}\frac{\overline{|A_{ij}^0|}}{\eta_0 B}, \frac{A_{ii}^0}{\eta_0 B} \text{ for } i = 1, \cdots, d\right\}\rfloor$, $A^K$ is still a Markov matrix.*

*Proof.* See more details in Appendix E.3 $\square$

*Remark* 4.6. According to Theorem 4.5, we can conclude that during the stage of fine-tuning on the specific downstream RL tasks, *Markov heads* preserve and they can never be switched to *Non-Markov heads*.

### 4.1.2. QUANTITATIVE ANALYSIS

In this section, we conduct quantitative analysis on the weight matrices transferred from PLMs to examine the existence of *Markov heads* in PLMs and their properties.

First, we measure the importance of each attention head. We initialize DT with pretrained GPT2-small parameters (`GPT2-DT`) and fine-tune it on RL tasks (Reid et al., 2022; Shi et al., 2024). Then, we test `GPT2-DT` under online environments, obtaining the predicted action $\tilde{a}$. To obtain the importance score of each head, we replace the output of the head $i$ with a zero vector (Bau et al., 2020; Michel et al., 2019) and notate the corresponding predicted action after removing head $i$ as $\tilde{a}_{-i}$. We define the importance score of the head $i$ as $\|\tilde{a} - \tilde{a}_{-i}\|_2$. The results are visualized in in Figure 2.

It is notable that some heads exhibit higher importance scores than the others. To further explore the inner property of these heads, we focus our analysis on the weight matrix

*Table 1.* We analyze the weight matrix $W_i^q(W_i^k)^T$ of each head $i$ in `GPT2` without fine-tuning (Initialization) and after fine-tuning (Fine-tuned). Details of (i) and (ii) can be found in Definition 4.1.

| Head Index | Initialization | | Fine-tuned | |
|:---:|:---:|:---:|:---:|:---:|
| | (i) | (ii) | (i) | (ii) |
| 0 | × | × | × | × |
| **1** | ✓ | ✓ | ✓ | ✓ |
| 2 | × | × | × | × |
| 3 | ✓ | × | × | × |
| 4 | ✓ | × | × | × |
| **5** | ✓ | ✓ | ✓ | ✓ |
| 6 | × | × | × | × |
| 7 | × | × | × | × |
| 8 | × | × | × | × |
| 9 | × | × | × | × |
| **10** | ✓ | ✓ | ✓ | ✓ |
| 11 | × | × | × | × |

$W_i^q(W_i^k)^T$ of each head $i$. From the statistical data in Table 1 and the heatmaps visualized in Figure 3, we can conclude that these heads with higher importance scores are exact *Markov heads*. Indeed, *Markov heads* exist in both PLMs and `GPT2-DT`, and such heads play a significant role on next-action prediction tasks.

*Remark* 4.7. Let $m$ and $n$ denote the mean of the absolute diagonal and off-diagonal elements in Definition 4.1. After applying the softmax function, the attention weight on the diagonal becomes $\frac{e^m}{e^m+(d-1)e^n}$, where $d$ is the embedding dimension. To determine a reasonable range for $r$, we assume $\frac{e^m}{e^m+(d-1)e^n}$ should be at least 0.5 to ensure that the head focuses more on the most recent input token. Solving this inequality yields $r > ln(d-1)+1$. In our setup, the embedding dimension $d=64$ and we set $r=8$ to identify *Markov heads* in `GPT2-DT`.

To verify both Theorem 4.3 and Theorem 4.5, we compare the attention score distributions before and after fine-tuning of both *Markov heads* and *Non-Markov heads*. Figure 4 visualizes the comparison with heatmaps. On the one hand, we observe that the attention distribution of *Markov heads* prior to fine-tuning is consistent with Theorem 4.3. Even when the embedding vectors for states, actions, and returns-to-go are randomly initialized, *Markov heads* exhibit strong attention to the last input token. This aligns with our theoretical result that $\mathbb{E}[\mathbf{e}_i A \mathbf{e}_i^T] \gg \mathbb{E}[\mathbf{e}_i A \mathbf{e}_j^T]$, where the last token corresponds to the current state $s_t$. On the other hand, we find that after fine-tuning, the weight product $W_i^q(W_i^k)^T$ for all *Markov heads* in `GPT2-DT` continues to satisfy the definition of a *Markov matrix*, providing empirical support for Theorem 4.5.

*Remark* 4.8. By Theorem 4.3 , we show that under any

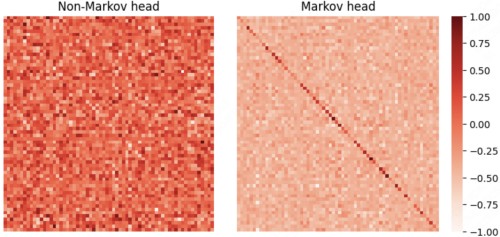

*Figure 3.* The heatmaps of $W_i^q(W_i^k)^T$ in *Non-Markov head* (left) and *Markov head* (right).

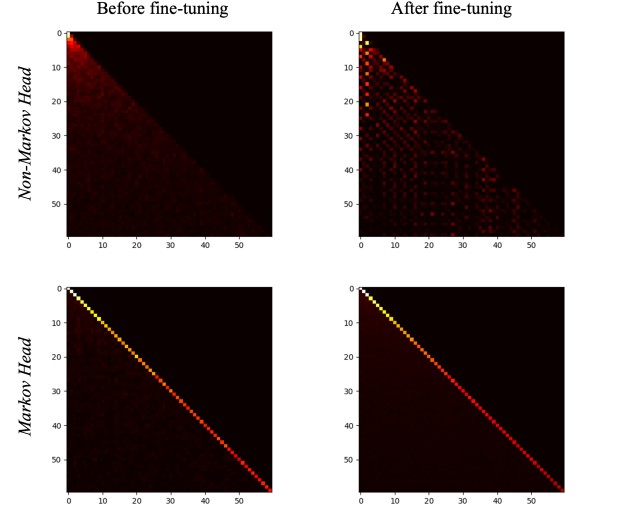

*Figure 4.* The two sub-figures above / below show the attention score matrices for *Non-Markov head* / *Markov head* before (left) and after fine-tuning (right). Lighter colors represent higher attention scores, while darker colors represent lower scores.

random embedding matrix, the property of extreme attention to the last input token preserves in expectation. Figure 4 further confirms our theorems that the property of *Markov Heads* holds before and after fine-tuning.

*Remark* 4.9. During the fine-tuning process, we set our learning rate as 1e-4 and used 10K steps for warming up, so the upper bound $\eta_0$ is approximately 1e-4. We observe that the magnitude of the maximum norm gradient of $W^q(W^k)^T$ is on the order of 1e-6, while the average absolute value of the off-diagonal elements is on the order of 1e-2. Given our setting of $r=8$, these observations indicate that the assumptions required by Theorem 4.5 are easy to satisfy under standard hyperparameter configurations.

## 4.2. `GPT2-DTMA`: A General Approach with Adaptive Planning Ability

Our analysis of *Markov heads* in `GPT2-DT` reveals that their extreme attention to the last-input token aligns closely

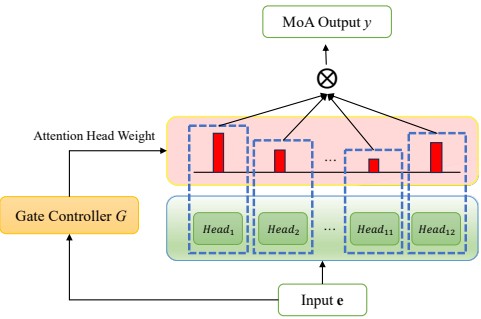

MoA Output $y$

Attention Head Weight

Gate Controller $G$

$Head_1$ $Head_2$ $\cdots$ $Head_{11}$ $Head_{12}$

Input $\mathbf{e}$

*Figure 5.* Mixture of Attention heads (MoA) allocates different weights to each attention head through a gate controller network.

with the behavior of memoryless policies (Littman, 1994).

However, such behavior may limit performance in *long-term* environments, where past trajectories carry rich contextual information that is crucial for accurate action prediction. In addition, it is often unclear whether long-term or short-term planning is required without prior knowledge of the environment. To address this challenge, we extend GPT2-DT and propose a general framework with adaptive planning ability, GPT2-DTMA, to handle both *short-term* and *long-term* environments adaptively.

From the results in Table 2, we observe that GPT2-DT performs well in MuJoCo Locomotion (*short-term*) environments, but exhibits a significant performance drop in Point-Maze (*long-term*) environments, where it underperforms compared to the baseline DT. This suggests that the effectiveness of *Markov heads* is inherently biased toward environments with specific planning ability requirements. This raises a natural question: How can we improve GPT2-DT so that it retains its advantages in *short-term* scenarios while closing the performance gap in *long-term* environments?

To address this key challenge, a natural idea is to adaptively control the influence of *Markov heads* based on the implicit requirement for planning ability of each environment. While the feed-forward network (FFN) following the attention layers is capable of integrating information, our analysis in Appendix F shows that it fails to effectively balance the contributions of *Markov* and *Non-Markov heads*. To overcome this limitation, we propose enhancing GPT2-DT with a Mixture of Attention (MoA) mechanism (Zhang et al., 2022; Jin et al., 2024), which results in our method named GPT2-DTMA. An illustration of the proposed architecture is shown in Figure 5. MoA treats each attention head as an expert specialized in capturing certain features or temporal dependencies. A trainable gating network $G_i$ is introduced to adaptively assign weights to each head $i$'s output. Given an input embedding vector $\mathbf{e}$, MoA computes a weighted sum over all attention head outputs, where the weights are dynamically determined by $G_i$. Formally, the MoA output

*Table 2.* We compare DT and GPT2-DT with mean normalized score ($\uparrow$) in MuJoCo Locomotion and mean test episode length ($\downarrow$) in PointMaze respectively. Dataset names are abbreviated as follows: medium as *m*, medium-replay as *m-r*.

| Environment | Dataset | DT | GPT-DT |
|---|---|---|---|
| MuJoCo Locomotion | Hopper-*m* | 67.4 | **77.9** |
| | Hopper-*m-r* | 74.1 | **77.9** |
| | Walker2d-*m* | 74.3 | **77.1** |
| | Walker2d-*m-r* | 71.9 | **74.0** |
| PointMaze | umaze | **56.3** | 59.7 |
| | medium | **158.7** | 188.0 |
| | large | **195.3** | 257.3 |

can be expressed as:

$$y(\mathbf{e}) = \sum_{i=1}^{N} G_i(\mathbf{e}) \cdot \text{Head}_i(\mathbf{e}). \quad (6)$$

The dense gate controller network $G$ is implemented as a linear layer $W_g$ followed by a softmax activation:

$$G_i(\mathbf{e}) = \text{softmax}_i(W_g(\mathbf{e})), \quad (7)$$

where $G_i(\mathbf{e})$ is the trainbale weight assigned to head $i$ and $\text{Head}_i(\mathbf{e})$ denotes the output of attention head $i$ with query projection $W_i^q$, keyword projection $W_i^k$, and value projection $W_i^v$. $\text{Head}_i(\mathbf{e})$ is defined as:

$$\text{Head}_i(\mathbf{e}) = \text{softmax}(\frac{(\mathbf{e}W_i^q)(EW_i^k)^T}{\sqrt{d_k}})(EW_i^v), \quad (8)$$

where $d_k$ denotes the dimension of the embedding layer.

## 5. Experiments

From the previous analysis, we identified the existence of *Markov head* in both PLMs and GPT2-DT and showed that they assign higher attention weights to the current state. However, it remains unclear whether this property benefits all reinforcement learning tasks. Prior studies (Reid et al., 2022; Shi et al., 2024; Yang & Xu, 2024) evaluate PLM-initialized decision transformers mainly on *short-term* environments such as MuJoCo Locomotion and Atari. In contrast, we aim to investigate whether *Markov heads* also contribute to next-action prediction in *long-term* environments. To this end, we adopt LaMo (Shi et al., 2024), a representative PLM-based decision transformer, as the implementation of our baseline GPT2-DT, and refer to it as such throughout the rest of the paper. We then compare our proposed method, GPT2-DTMA, with several baselines, including the standard Decision Transformer (DT) (Chen et al., 2021), the value-based offline RL algorithm CQL

*Table 3.* Mean normalized score (↑) with three seeds in MuJoCo Locomotion (*short-term* environment). The dataset names are abbreviated as follows: medium as *m*, medium-replay as *m-r*, medium-expert as *m-e* and Halfcht as Halfcheetah.

| Dataset | CQL | DT | DC | DTMA | GPT2-DT | GPT2-DTMA | GPT2-P |
|---|---|---|---|---|---|---|---|
| Hopper-m | $57.7_{\pm1.6}$ | $67.4_{\pm1.7}$ | $\mathbf{79.7}_{\pm1.1}$ | $65.2_{\pm2.3}$ | $77.9_{\pm1.1}$ | $77.4_{\pm0.8}$ | $70.8_{\pm1.4}$ |
| Hopper-m-r | $73.7_{\pm4.4}$ | $74.1_{\pm4.1}$ | $\mathbf{82.3}_{\pm2.5}$ | $74.7_{\pm3.7}$ | $77.9_{\pm2.1}$ | $80.4_{\pm1.8}$ | $75.3_{\pm2.9}$ |
| Hopper-m-e | $107.3_{\pm0.4}$ | $108.6_{\pm0.3}$ | $111.4_{\pm0.2}$ | $109.4_{\pm0.1}$ | $111.7_{\pm0.1}$ | $111.6_{\pm0.1}$ | $109.4_{\pm0.2}$ |
| Walker2d-m | $72.6_{\pm1.3}$ | $74.3_{\pm1.9}$ | $79.3_{\pm1.3}$ | $74.0_{\pm1.4}$ | $77.1_{\pm1.1}$ | $\mathbf{79.9}_{\pm0.8}$ | $73.3_{\pm2.7}$ |
| Walker2d-m-r | $78.3_{\pm2.2}$ | $71.9_{\pm2.7}$ | $\mathbf{79.2}_{\pm1.8}$ | $70.5_{\pm3.4}$ | $74.0_{\pm2.9}$ | $77.0_{\pm2.5}$ | $71.5_{\pm3.5}$ |
| Walker2d-m-e | $107.8_{\pm0.2}$ | $107.6_{\pm0.1}$ | $108.3_{\pm0.2}$ | $108.1_{\pm0.2}$ | $108.1_{\pm0.1}$ | $\mathbf{108.6}_{\pm0.1}$ | $108.2_{\pm0.1}$ |
| Halfcheetah-m | $43.1_{\pm0.3}$ | $42.8_{\pm0.4}$ | $43.1_{\pm0.2}$ | $43.5_{\pm0.3}$ | $42.6_{\pm0.5}$ | $43.0_{\pm0.4}$ | $42.3_{\pm0.6}$ |
| Halfcheetah-m-r | $\mathbf{42.4}_{\pm1.9}$ | $39.2_{\pm2.4}$ | $38.3_{\pm3.4}$ | $38.8_{\pm3.3}$ | $39.8_{\pm3.1}$ | $40.3_{\pm2.9}$ | $39.6_{\pm3.1}$ |
| Halfcheetah-m-e | $88.4_{\pm1.1}$ | $91.9_{\pm1.3}$ | $92.1_{\pm1.3}$ | $91.8_{\pm1.1}$ | $92.3_{\pm0.7}$ | $\mathbf{92.5}_{\pm1.1}$ | $92.2_{\pm0.5}$ |

*Table 4.* Episode length (↓) in different size of Maze (*long-term* environment).

| Environment | Dataset | CQL | DT | DC | DTMA | GPT2-DT | GPT2-DTMA |
|---|---|---|---|---|---|---|---|
| | umaze | $58.0_{\pm1.0}$ | $56.3_{\pm1.7}$ | $\mathbf{54.3}_{\pm2.3}$ | $58.7_{\pm1.3}$ | $59.0_{\pm3.0}$ | $55.0_{\pm2.0}$ |
| PointMaze | medium | $\mathbf{134.0}_{\pm8.0}$ | $158.7_{\pm7.3}$ | $236.3_{\pm6.7}$ | $155.0_{\pm8.0}$ | $188.0_{\pm13.0}$ | $146.3_{\pm9.3}$ |
| | large | $\mathbf{162.0}_{\pm6.0}$ | $195.3_{\pm14.7}$ | $288.0_{\pm12.0}$ | $217.0_{\pm16.0}$ | $257.3_{\pm13.3}$ | $203.0_{\pm11.0}$ |

(Kumar et al., 2020), and Decision Convformer (DC) (Kim et al., 2023). Full experimental setup details are provided in Appendix C.

### 5.1. *Short-term* and *Long-term* Environments

In this section, we provide brief introduction to each experimental task and categorize them into *short-term* and *long-term* environments, as follows:

- MuJoCo Locomotion tasks (Fu et al., 2020) are physics engines designed primarily for simulating and controlling articulated rigid body systems. Kim et al. (2023) believe that *short-term* planning ability is enough for achieving a good performance in those tested environments. If a policy can control the agent to reach further or move faster while maintaining stable posture, then the normalized score will be higher.

- In a Maze task, such as PointMaze, a policy model is trained to direct the object from a random start position to the goal in a maze. Note that this point mass can only observe its own position and the position of the goal; it has no knowledge of the positions of obstacles in the maze. The policy model needs to learn how to outline the entire maze based on past trajectory information. Therefore, this environment is generally considered to require model's *long-term* planning ability (Lin et al., 2022). We evaluate whether the policy model can guide the point mass to the goal using a shorter episode length as the evaluation metric.

### 5.2. Main Results

To validate the effectiveness of MoA in regulating the attention heads of GPT2-DT, we evaluate the performance of GPT2-DTMA proposed in Section 4.2 across both short-term and long-term environments. As shown in Table 3, GPT2-DTMA outperforms DT in most *short-term* environments. Notably, MoA introduces only a modest increase in model complexity—approximately a 3% rise in parameter count over the base DT—making it a lightweight yet effective enhancement.

In *long-term* environments, Table 4 shows that integrating the MoA architecture into GPT2-DT enables the agent to reach the goal in PointMaze with shorter episode lengths. However, as stated in Theorem 4.5, MoA can only regulate the influence of *Markov heads*, whose inherent properties are preserved through fine-tuning. Consequently, due to the adverse impact of *Markov heads* in long-term scenarios, GPT2-DTMA can reduce the performance gap with DT while suffering difficulty to outperform it. Detailed trends in how MoA adjusts the weights of *Markov* and *Non-Markov* heads across short-term and long-term environments are provided in Appendix G.

By comparing the performance of the same model in Table 3 and Table 4, we observe that both DC and CQL exhibit strong performance only in specific scenarios that align with their respective planning ability biases, but struggle in other settings. In contrast, our model GPT2-DTMA with adaptive planning ability maintains relatively consistent and competitive performance across both *short-term* and *long-*

*Table 5.* GPT2-DTMA with different context length ($k$) in PointMaze-large. $G_{\text{Markov}}$ represents the sum of weights for all *Markov heads*. $R_{\text{Markov}}$ represents the percentage of $G_{\text{Markov}}$ relative to that when $k = 10$.

| $k$ | Episode length ($\downarrow$) | $G_{\text{Markov}}$ | $R_{\text{Markov}}$ |
|-----|---|---|---|
| $k = 10$ | 236.3 | 0.397 | 100.0% |
| $k = 20$ | 230.7 | 0.310 | 78.1% |
| $k = 30$ | 226.0 | 0.234 | 58.9% |
| $k = 40$ | 214.3 | 0.188 | 47.3% |
| $k = 50$ | 203.0 | 0.167 | 42.1% |

*Table 6.* GPT2-DTMA performance using different context length ($k$) in Hopper-medium environment. $G_{\text{Markov}}$ represents the sum of weights for all *Markov heads*. $R_{\text{Markov}}$ represents the percentage of $G_{\text{Markov}}$ relative to that when $k = 10$.

| $k$ | Normalized score ($\uparrow$) | $G_{\text{Markov}}$ | $R_{\text{Markov}}$ |
|-----|---|---|---|
| $k = 10$ | 77.1 | 0.505 | 100.0% |
| $k = 20$ | 77.4 | 0.488 | 96.6% |
| $k = 30$ | 76.3 | 0.480 | 95.0% |
| $k = 40$ | 77.0 | 0.461 | 91.3% |
| $k = 50$ | 75.8 | 0.440 | 87.1% |

*term* environments.

Furthermore, we conduct an ablation study by incorporating the MoA mechanism into a randomly initialized DT, referred to as DTMA. Table 3 shows that this method does not lead to performance gains, as evidenced by the comparable results between DTMA and DT. This suggests that the effectiveness of MoA primarily lies in its ability to modulate the influence of *Markov heads* when predicting the next action, a property not present in models without pretraining.

### 5.3. Attention Score Distribution of GPT2-DTMA in Different Environments

In this section, we investigate whether GPT2-DTMA is capable of adaptively adjusting the weights of *Markov heads* to better adapt to different environments. We use the most challenging task, PointMaze-large, as our testbed, which is a typical long-term environment. In *long-term* environments, models are required to extract useful information from past sequences, rather than relying primarily on current state. However, the *Markov heads* tend to focus attention disproportionately on the most recent input, limiting the model's ability to leverage long-range context and thereby negatively affecting performance. By using MoA to reduce the influence of *Markov heads*, GPT2-DTMA can alleviate their negative impact on action predictions. As shown in Table 5, it can be seen that as the length of context length increases, both the sum of weight $G_{\text{Markov}}$ assigned to *Markov heads* and the number of steps required for the point mass to reach the goal gradually decrease. This suggests that GPT2-DTMA endeavors to capture previous information that is further away from the current state.

Similarly, we conducted a comparative experiment examining the effect of context length in *short-term* environments. As context length increases, the model shows a growing tendency to attend to distant information, leading to an overall decrease in $G_{\text{Markov}}$, as shown in Table 6. However, at the same context length, the $R_{\text{Markov}}$ in *short-term* environments is significantly higher compared to those in *long-term* environments. This suggests that, in *short-term* environments,

MoA allows *Markov heads* to retain a higher influence in action prediction, whereas in *long-term* environments, the model relies more heavily on other attention heads. These findings indicate that MoA can effectively identify whether the environment requires *short-term* or *long-term* planning ability and adaptively modulate weights across different heads to suit the environment.

## 6. Discussion

To further validate the critical role of *Markov head* in *short-term* environments, we design two different approaches: The first approach explicitly reduces the influence of *Markov heads* by lowering their MoA weights. The second approach modifies the initialization of the GPT2-DT model, using parameters that do not satisfy the *Markov matrix* property.

### 6.1. Reducing the Weights of *Markov Heads* in Action Prediction

To assess the role of *Markov heads* in short-term environments, we conduct a comparative experiment against GPT2-DTMA by intentionally reducing their contribution to action prediction. To this end, we develop a variant model called GPT2-P, which introduces a penalty on the weights of *Markov heads* in the loss function:

$$\mathcal{L}_{\text{GPT2-P}} = L_{\text{DT}} + \alpha \cdot \sum_{i \in [1,5,10]} G_i(x), \quad (9)$$

where $[1, 5, 10]$ are the indices of *Markov heads*.

By applying a penalty coefficient of $\alpha = 0.1$, the cumulative weight assigned to the *Markov heads* is reduced from 0.488 in GPT2-DTMA to 0.184 in its variant, GPT2-P. As shown in Table 3, GPT2-P fails to match the policy performance of GPT2-DTMA in short-term tasks. This degradation underscores the significance of *Markov heads* in improving action selection under limited planning ability. In many cases, reducing their contribution causes GPT2-P to regress to the performance of the standard DT baseline. Furthermore, these results suggest that other components pretrained for

*Table 7.* Mean normalized score (↑) of DT initialized with different pretrained parameters in MuJoCo Locomotion.

| Dataset | DT | GPT2-DT | CLIP-DT | GPTJ-DT |
|---|---|---|---|---|
| Hopper-m | $67.4_{\pm 1.7}$ | $77.9_{\pm 1.1}$ | $66.9_{\pm 1.3}$ | $71.6_{\pm 1.5}$ |
| Hopper-m-r | $74.1_{\pm 4.1}$ | $77.9_{\pm 2.1}$ | $76.1_{\pm 2.5}$ | $72.9_{\pm 1.7}$ |
| Hopper-m-e | $108.6_{\pm 0.3}$ | $111.7_{\pm 0.1}$ | $109.2_{\pm 0.2}$ | $109.6_{\pm 0.1}$ |
| Walker2d-m | $74.3_{\pm 1.9}$ | $77.1_{\pm 1.1}$ | $74.2_{\pm 0.8}$ | $74.8_{\pm 1.2}$ |
| Walker2d-m-r | $71.9_{\pm 2.7}$ | $74.0_{\pm 2.9}$ | $70.3_{\pm 4.1}$ | $70.4_{\pm 2.3}$ |
| Walker2d-m-e | $107.6_{\pm 0.1}$ | $108.1_{\pm 0.1}$ | $107.9_{\pm 0.3}$ | $108.1_{\pm 0.1}$ |
| Halfcheetah-m | $42.8_{\pm 0.4}$ | $42.6_{\pm 0.5}$ | $42.6_{\pm 0.6}$ | $42.9_{\pm 0.3}$ |
| Halfcheetah-m-r | $39.2_{\pm 2.4}$ | $39.8_{\pm 3.1}$ | $39.8_{\pm 3.3}$ | $39.3_{\pm 2.5}$ |
| Halfcheetah-m-e | $91.9_{\pm 1.3}$ | $92.3_{\pm 0.7}$ | $92.1_{\pm 0.9}$ | $92.1_{\pm 0.9}$ |

NLP tasks—such as the feed-forward network (FFN) layers—may offer limited transferability to the reinforcement learning domain.

### 6.2. Initializing DT Using Pretrained Parameters without *Markov Heads*

In our offline RL setup with an autoregressive formulation, we initialize the model using pretrained text encoders from CLIP and GPT-J. While CLIP includes both text and image encoders, we adopt only its text encoder, which is an autoregressive Transformer pretrained on large-scale textual data. CLIP is trained to align textual descriptions with corresponding images, whereas GPT-J is a GPT2-style causal language model, but pretrained on a different dataset (Pile) rather than the WebText corpus used by GPT2. Despite architectural similarity and pretraining on textual data, we find that the attention layers in both CLIP and GPT-J do not exhibit the *Markov matrix* structure observed in our GPT2-DTMA model. We attribute this to differences in training objectives: unlike GPT2, whose next-token prediction loss may encourage local temporal dependencies, the contrastive objective in CLIP and the different pretraining distribution of GPT-J do not appear to induce the same structural bias in the attention heads.

As shown in Table 7, initializing DT with pretrained parameters from CLIP (CLIP-DT) or GPT-J (GPTJ-DT) does not yield performance comparable to GPT2-DT. The presence of *Markov heads* in GPT2-DT enables the model to allocate sufficient attention to the current state, resulting in improved performance over the vanilla DT. The absence of the *Markov matrix* structure in CLIP-DT is expected, given its contrastive training objective. However, although GPT-J is pretrained using the same next-token prediction loss as GPT2, it fails to exhibit this property. This discrepancy may be attributed to differences in the pretraining data distribution and scale.

## 7. Conclusion

In this paper, we addressed a key question: how do pretrained language models (PLMs) influence fine-tuning performance in offline reinforcement learning? We introduced the concept of *Markov heads* and investigated their distinctive properties. Our analysis reveals that *Markov heads* are the primary transferable inductive bias from PLMs, beneficial in *short-term* tasks but detrimental in *long-term* scenarios. To address this challenge, we proposed GPT2-DTMA, a general and adaptive approach that enables the model to automatically adjust its planning capacity across different environments. Experimental results demonstrate that GPT2-DTMA significantly narrows the performance gap in both short- and long-term settings. We believe that understanding the mechanisms behind PLM transfer opens promising directions for future research, such as optimizing the use of pretrained parameters in RL or designing pretraining objectives tailored for decision-making tasks.

## Acknowledgements

This work is partially supported by NSFC (61732006) and Public Computing Cloud, Renmin University of China.

## Impact Statement

This paper presents work whose goal is to advance the field of pretrained decision transformer. There are many potential societal consequences of our work, none which we feel must be specifically highlighted here.

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

## A. Pretrained Language Models for Offline RL Tasks

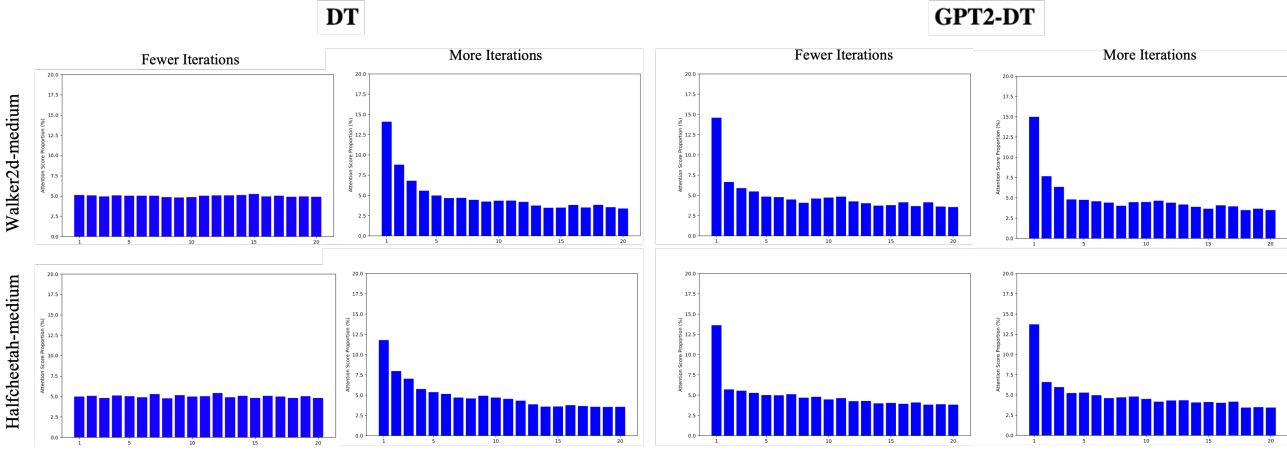

*Figure 6.* The paradigm for transferring PLM parameters to offline RL tasks.

Figure 6 follows the paradigm for transferring knowledge from the natural language domain to the RL domain in previous work (Reid et al., 2022). In the first stage, we pretrain Transformer with language data to predict the next token. In the second stage, we fine-tune pretrained Transformer on downstream offline RL tasks to predict the next actions.

## B. Comparisons of Attention Score Distributions

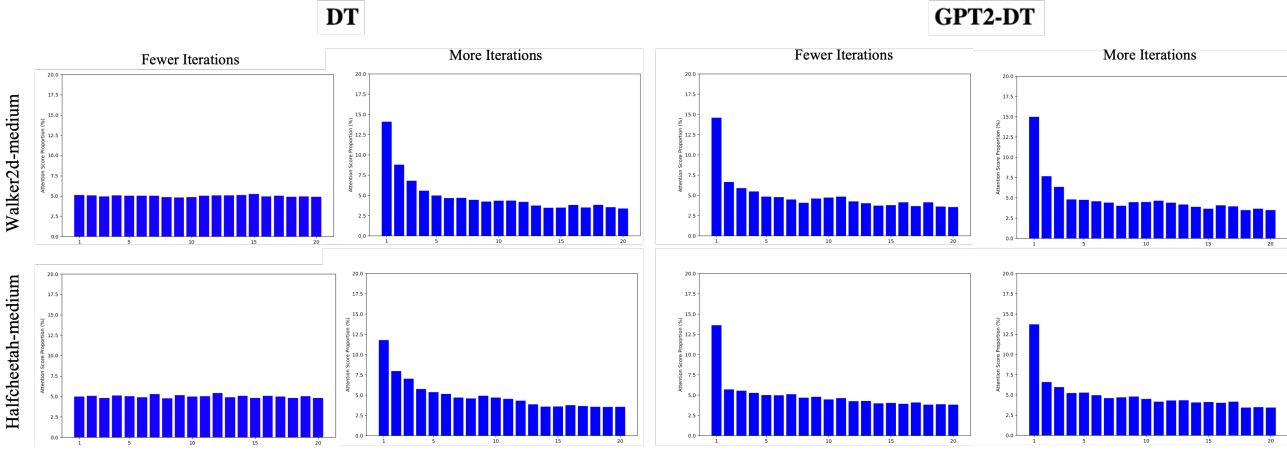

*Figure 7.* We compare the attention score distribution of `DT` and `GPT2-DT` during the process of fine-tuning within fewer iterations and more iterations in Walker2d-medium (upper row) and Halfcheetah-medium (lower row) environments.

We explore how the attention score distribution differs between `DT` and `GPT2-DT` when fine-tuning with a smaller number of iterations versus a larger number of iterations. Figure 7 illustrates that in different environments, `GPT2-DT` can still learn the ability to focus on closer positional information in fewer training iterations compared to `DT`. This ability to concentrate on recent information is precisely what is needed to excel in short-term environments.

## C. Experiment Setup

We use GPT2-small pretrained parameters to initialize `DT` transformer layer. To fine-tune pretrained language models, we set the learning rate as 1e-4 with 10K warmup steps and weight decay as 1e-4. We fine-tune the model with 100K steps and batch size as 64. Our training process uses a single Nividia A100 and four Tesla V100 with cuda version of 12.3. During training, the agent is evaluated every 500 steps, by running 20 episodes. We report results averaged among 3 seeds (seeds $0, 1, 2$ are used). The offline datasets used for training the model are collected in D4RL (Fu et al., 2020) benchmark. For MuJoCo Locomotion tasks, the tasks we selected are *halfcheetah, hopper* and *walker2d*. Each task has 3 datasets collected

with different strategies: *medium* dataset consisting of 1M interaction samples of a middle-level RL agent. *medium-replay* includes the whole replay buffer when training an RL agent until it reaches middle-level performance. *medium-expert* is a mixture of the *medium* dataset and 1M interaction samples of an expert-level RL policy. For Maze tasks, *umaze*, *medium* and *large* represent square mazes with side lengths of 5, 8, and 12, respectively.

## D. Influence of *Markov heads* in *Long-term* Environments

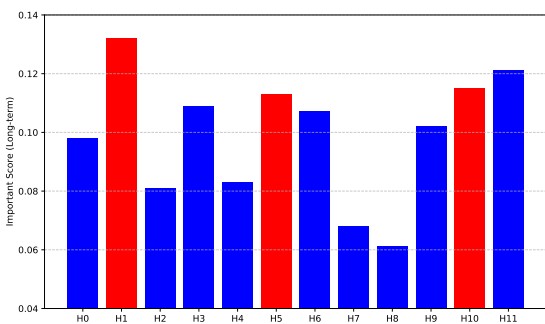

*Figure 8.* Importance scores on different heads in PointMaze (*long-term* environment). The red bar represents the importance scores allocated to the *Markov head* in the first attention layer.

Figure 8 shows the importance scores of different attention heads when predicting actions. Similar to the phenomenon in *short-term* environments, the *Markov heads* exhibit large influences in predicting actions. It indicates that the poor performance of GPT2-DT in *long-term* environments may stem from the *Markov heads* focusing only on the current state.

## E. Theoretical Proofs

In this section, we present proofs of Theorem 4.3, Corollary 4.4 and Theorem 4.5.

### E.1. Proof of Theorem 4.3

We will give a simple proof showing that *Markov matrix*, i.e., $W_i^q W_i^{k^T}$ satisfies Definition 4.1, will pay most attention on the last-input token under random embedding initializations, i.e., $\mathbb{E}\left[E(W_i^q W_i^{k^T})E^T\right]$ also satisfies Definition 4.1, where $E \in \mathbb{R}^{K \times d}$ is a random embedding matrix, $K$ is the sequence length and in this case, $A = \mathbb{E}\left[E(W_i^q W_i^{k^T})E^T\right]$.

*Proof.* For any $\mathbf{e}_m = (e_m^1, e_m^2, \cdots, e_m^d)$ and $\mathbf{e}_n = (e_n^1, e_n^2, \cdots, e_n^d)$, where $m, n \in \{1, \cdots, K\}$ and $m \neq n$, we have

$$
\begin{aligned}
\mathbb{E}[\mathbf{e}_m A \mathbf{e}_n^T] &= \sum_{i=1}^d \sum_{j=1}^d A_{ij} \mathbb{E}[e_m^i e_n^j] \\
&= \sum_{i=1}^d A_{i,i} \mathbb{E}[e_m^i e_n^i] + \sum_{i,j=1,\cdots,d, i \neq j} A_{i,j} \mathbb{E}[e_m^i e_n^j] \\
&= \sum_{i=1}^d A_{i,i} \mathbb{E}[e_m^i e_n^i] + \left(\mathbb{E}[e_m^i]\right)^2 \sum_{i,j=1,\cdots,d, i \neq j} A_{ij} \\
&= \sum_{i=1}^d A_{i,i} \mathbb{E}[e_m^i e_n^i] = 0,
\end{aligned}
\tag{10}
$$

where the last equality is because $\mathbb{E}[e_m^i] = \mathbb{E}[e_n^i] = 0$. And we know that $e_m^i$ and $e_n^i$ are i.i.d, hence by taking the

expectation, we can obtain that

$$\mathbb{E}[e_m^i e_n^i] = \mathbb{E}[e_m^i]\mathbb{E}[e_n^i] = \left(\mathbb{E}[e_m^i]\right)^2 < \mathbb{E}\left[(e_m^i)^2\right], \tag{11}$$

where the last inequality is due to $Var[e_m^i] = \mathbb{E}\left[(e_m^i)^2\right] - \left(\mathbb{E}[e_m^i]\right)^2 > 0$. Also, by simply replacing $\mathbf{e}_n^T$ with $\mathbf{e}_m^T$ in Equation (10), we have

$$\begin{aligned}
\mathbb{E}[\mathbf{e}_m A \mathbf{e}_m^T] &= \sum_{i=1}^{d} A_{i,i}\mathbb{E}\left[(e_m^i)^2\right] + \left(\mathbb{E}[e_m^i]\right)^2 \sum_{i,j=1,\cdots,d,i\neq j} A_{ij} \\
&= \sum_{i=1}^{d} A_{i,i}\mathbb{E}\left[(e_m^i)^2\right] = \sum_{i=1}^{d} A_{i,i}.
\end{aligned} \tag{12}$$

Since we know that $\{A_{i,i}\}_{i=1}^{d}$ are all positive, we can conclude that $\mathbb{E}(\mathbf{e}_m A \mathbf{e}_m^T) > \mathbb{E}(\mathbf{e}_m A \mathbf{e}_n^T) = 0$. Therefore, for any $r > 0$, Definition 4.1 will be satisfied. $\square$

### E.2. Proof of Corollary 4.4

*Proof.* By Theorem 4.3, we know that $\frac{\mathbb{E}[D]}{\mathbb{E}[O]} > r$ for some $r > 0$. And for any $\varepsilon \triangleq \frac{\mathbb{E}[D]-r\mathbb{E}[O]}{2r} > 0$, we have $\frac{\mathbb{E}[D]-\varepsilon}{\mathbb{E}[O]+\varepsilon} > r$. We apply concentration bounds for sub-exponential random variables, noting that $|\Pi_{ij}|$ is sub-exponential with sub-Gaussian proxy $\|A\|_F$. For the diagonal mean $D$, there are $K$ i.i.d. terms, and for the off-diagonal mean $O$, there are $K(K-1)$ terms. By Bernstein inequality and Hanson-Wright inequality, we obtain:

$$\mathbb{P}\left(|D - \mathbb{E}[D]| \geq \varepsilon\right) \leq 2\exp\left(-c_1 K \min\left(\frac{\varepsilon^2}{\|A\|_F^2}, \frac{\varepsilon}{\|A\|}\right)\right), \tag{13}$$

and

$$\mathbb{P}\left(|O - \mathbb{E}[O]| \geq \varepsilon\right) \leq 2\exp\left(-c_2 K(K-1) \min\left(\frac{\varepsilon^2}{\|A\|_F^2}, \frac{\varepsilon}{\|A\|}\right)\right). \tag{14}$$

Taking union bound and noting that if $D \geq \mathbb{E}[D] - \varepsilon$ and $O \leq \mathbb{E}[O] + \varepsilon$, then

$$\frac{D}{O} > \frac{\mathbb{E}[D] - \varepsilon}{\mathbb{E}[O] + \varepsilon} > r. \tag{15}$$

Then we have

$$\mathbb{P}(\frac{D}{O} \leq r) \leq \mathbb{P}(D < \mathbb{E}[D] - \varepsilon) + \mathbb{P}(O > \mathbb{E}[O] + \varepsilon) \leq \delta_0, \tag{16}$$

where $\delta_0 \triangleq 2\exp\left(-c_1 K \min\left(\frac{\varepsilon^2}{\|A\|_F^2}, \frac{\varepsilon}{\|A\|}\right)\right) + 2\exp\left(-c_2 K(K-1)\min\left(\frac{\varepsilon^2}{\|A\|_F^2}, \frac{\varepsilon}{\|A\|}\right)\right)$. Therefore under the definition of $\varepsilon$, we obtain the desired high-probability bound. $\square$

### E.3. Proof of Theorem 4.5

In this section, we will discuss the phenomenon in Table. 2 and give a theoretical analysis for Theorem 4.5.

*Proof.* Suppose that the total number of training iterations is $K$, and $A^0 = W^q W^{k^T}$ is a *Markov matrix*, the difference between the initialized parameters $A^0$ and the trained parameters is denoted as $\Delta A^K = \sum_{k=1}^{K} \eta_k g_k$, where $\eta_k$ denotes the learning rate at the $k$-th iteration and $A^K = A^0 - \Delta A^K$. By choosing the Adam optimizer, we know that there exists $\eta_0 > 0$ such that $\eta_k \leq \eta_0$ for all $k = 1, \ldots, K$. Hence, we have

$$0_{d\times d} \leq |\Delta A^K| = |\sum_{k=1}^{K} \eta_k g_k| \leq \sum_{k=1}^{K} \eta_k |g_k| \leq \eta_0 K B \cdot \mathbf{1}_{d\times d}. \tag{17}$$

Let $a_{max} = \max\{|\Delta A^K_{i,j}|, where\ i,j = 1, \cdots, d\}$ and $a_{max} \leq \eta_0 KB$. Due to Definition 4.1, we know that there exists $\rho > 0$, such that $\frac{\overline{A^0_{ii}}}{|A^0_{ij}|} \geq r + \rho > r$, i.e., $\overline{A^0_{ii}} \geq (r + \rho)\overline{|A^0_{ij}|}$. Since $a_{max} < \min_{i=1,\cdots,d} A^0_{ii}$, then we have

$$A^K_{ii} = A^0_{ii} - \Delta A^K_{ii} \geq A^0_{ii} - a_{max} > 0,$$
$$|A^0_{ij}| - a_{max} \leq |A^K_{ij}| = |A^0_{ij} - \Delta A^K_{ij}| \leq |A^0_{ij}| + a_{max}. \tag{18}$$

thus, $\overline{A^K_{ii}} \geq \overline{A^0_{ij}} - a_{max}$ and $\max\{\overline{|A^0_{ij}|} - a_{max}, 0\} \leq \overline{|A^K_{ij}|} \leq \overline{|A^0_{ij}|} + a_{max}$. Since $a_{\max} \leq \eta_0 KB < \frac{\rho}{r+1}\overline{|A^0_{ij}|}$, then $(r+1)a_{max} < \rho\overline{|A^0_{ij}|} < \overline{A^0_{ii}} - r \cdot \overline{|A^0_{ij}|}$, i.e.,

$$\frac{\overline{A^K_{ii}}}{|A^K_{ij}|} \geq \frac{\overline{A^0_{ii}} - a_{max}}{\overline{|A^0_{ij}|} + a_{max}} > r. \tag{19}$$

Hence, $A^K$ is still a *Markov matrix*.

$\square$

## F. FFN Cannot Achieve Adaptive Attention Facing Different Environments

| Dataset | GPT2-DT | GPT2-DT (part) | GPT2-DTMA |
|---------|---------|----------------|-----------|
| umaze   | 59      | 67             | **55**    |
| medium  | 188     | 229            | **146**   |
| large   | 257     | 286            | **203**   |

*Table 8.* Episode length ($\downarrow$) in different size of PointMaze (a *long-term* environment). `GPT2-DT` fine-tune all parameters in the model and `GPT2-DT` (part) only fine-tune the parameters in embedding layer and FFN layer.

The FFN layer can weight the output information from different attention heads, but unlike MoA, which assigns a shared importance score to the output of each attention head, the FFN applies a nonlinear transformation to the outputs of the attention heads. To verify the necessity of our proposed combining `GPT2-DT` with MoA, it can be observed from Table 8 that relying solely on the FFN layer is insufficient for effectively balancing the influences of the *Markov heads* and *Non-Markov heads* in action prediction.

## G. Changes of MoA Weight between *Markov Heads* and *Non-Markov Heads*

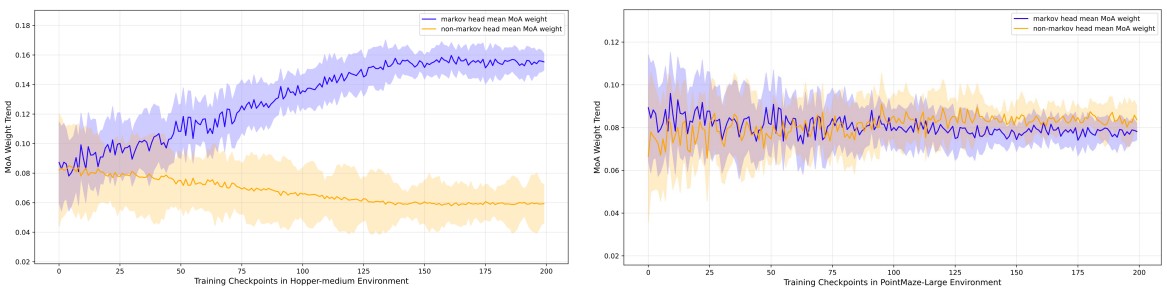

*Figure 9.* We illustrate the changes of MoA weight between *Markov heads* and *Non-Markov heads* in short-term environment (left) and long-term environment (right).

To better observe the differences in the impact of MoA in short-term and long-term environments, we show the MoA weights assigned for *Markov heads* and *Non-Markov heads* while training in Figure 9. We found that, after the MoA weights have converged, the *Markov heads* is more influential than the *Non-Markov heads* in predicting actions in short-term environments, while in long-term environments, the *Non-Markov heads* hold greater importance. This indicates that combining the MoA structure enables the model to identify whether the planning ability required by the environment is short-term or long-term.

