# OpenReview forum: "Unveiling Markov heads in Pretrained Language Models for Offline Reinforcement Learning"
_ICML.cc/2025/Conference — ICML 2025 poster_

### Official Review · Reviewer_L8Sr · 2025-03-01

**Overall Recommendation:** 4

**Summary:**

Previous works in the area of reinceforcement learning (RL)/foundation models have shown that pre-trained language models (PLMs) can enhance the performance of offline RL. This paper studies an important question: what kind of knowledge from PLMs has been transferred to RL to achieve such good results? They study the attention score distribution of decision-transformer (DT) and PLM-DT, and find that after training some iterations, the score distribution of DT is similar to the initial distribution of PLM-DT. They thus defined the concept of Markov heads, and further show that Markov heads are the key property transferred from PLM, both theoretically and empirically. Furthermore, they find that Markov heads are only beneficial for short-term environments, empirically. And they propose a general approach called GPT-DPMA, which show advantages over GPT-DT across both long-term/short-term tasks.

**Claims And Evidence:**

**claim 1**: the score distribution of DT is similar to the initial distribution of PLM-DT

evidence: Figure 1, DT is only tested on one task, not enough

**claim 2**: the markov heads will not change after fine-tuning

evidence: Figure 4, only tested on one task, not enough

**claim 3**: markov heads are beneficial for short-term tasks, while not for long-term tasks

evidence: Table 3, solid

**claim 4**: MoA can reduce the performance gap in long-term tasks

evidence: Table 4, solid

**Essential References Not Discussed:**

I don't think there is any work essentially related to the work not cited.

**Experimental Designs Or Analyses:**

I've checked the experimental parts. The experiments are conducted in a common way in this area. One issue is that some of the experiments are only conducted on one specific task, like hopper. More systematic experiments should be conducted.

**Methods And Evaluation Criteria:**

Yes, the proposed methods are practical, and the benchmark d4rl is commonly used.

**Other Comments Or Suggestions:**

no

**Other Strengths And Weaknesses:**

Strengths:
- The paper is well-written.
- The observation is novel.

Weakness:
- See my comments regarding to ``Claims And Evidence`` and ``Theoretical claims``.

**Questions For Authors:**

- On long-term tasks (Table 4), GPT-DTMA still cannot surpass DT. Could the author give some explanations?
- Would the positional embedding play a role in yielding Markov heads? For example, if the positional embedding is relative, then the model might be trained to attend to nearest token.

**Relation To Broader Scientific Literature:**

I think this paper is important for this area. Many prior works [1,2,3] show the power of PLMs for downstream low-level RL tasks, however, this paper would be the first to offer a reasonable explanation for this phenomenon. Therefore, it would be of significant value for the study of foundation models for decision-making, if their claims are further systematically supported.

[1] Unleashing the Power of Pre-trained Language Models for Offline Reinforcement Learning. Shi et al. ICLR 2024.

[2] Decomposed Prompt Decision Transformer for Efficient Unseen Task Generalization. Zheng et al. NeurIPS 2024.

[3] Pre-trained Language Models Improve the Few-shot Prompt Ability of Decision Transformer. Yang et al. arXiv 2024.

**Theoretical Claims:**

I've checked the proofs for all theoretical results (Theorem 4.3 and Theorem 4.5).

Though the proofs are correct, the results are not very strong. 1. The result that $\mathbb E[EAE^\top]$ is a markov matrix is not very useful. It would be better if the authors could prove that, w.h.p., $EAE^\top$ is a markov matrix. 2. The scale of $K$ is uncertain, and might be very small.

---

> ### Author Rebuttal · Authors · 2025-04-01
>
> Thank you for your detailed review and feedback. We appreciate your positive comments about the novelty and presentation of our work. Please kindly find the response to your concerns below.
>
> **W1. For figure 1 and figure 4, DT is tested on only one task.**
> We have tested DT and GPT-DT on more tasks and the results are shown in Fig.1~10 at [[URL](https://anonymous.4open.science/r/submission9905-7D09/readme.md)]. The conclusion are consistent among different tasks. We will include these results from multiple tasks in the final version.
>
> **W2. Stronger results for Theorem 4.3**
> We extend our proof to show high probability bound. By Theorem 4.3 (page 4), we know $\frac{\mathbb{E}[D]}{\mathbb{E}[O]} > r$ for some $r>0$, where $D \triangleq \frac{1}{K} \sum_{i=1}^K |\Pi_{ii}|,
> O \triangleq \frac{1}{K(K-1)} \sum_{i \ne j} |\Pi_{ij}|, {\rm and\ } \Pi\triangleq E A E^T.$ We assume each element in embedding vector is i.i.d sampled from $\mathcal{N}(0,1)$. Then  $|\Pi_{ij}|$  is sub-exponential with sub-Gaussian proxy $||A||_F$. By Bernstein inequality and Hanson-Wright inequality, we can get the bound for $\mathbb{P}\left( |D - \mathbb{E}[D]| \geq \varepsilon \right)$ and $\mathbb{P}\left( |O - \mathbb{E}[O]| \geq \varepsilon \right)$. By choosing $\varepsilon\triangleq \frac{\mathbb{E}[D] - r \mathbb{E}[O]}{2r} > 0$, we have $\frac{D}{O} > \frac{\mathbb{E}[D] - \varepsilon}{\mathbb{E}[O] + \varepsilon} > r$. Due to $\mathbb{P}(\frac{D}{O}\leq r) \leq \mathbb{P}(D < \mathbb{E}[D]-\varepsilon) + \mathbb{P}(O> \mathbb{E}[O] + \varepsilon) \leq \delta_0$, we can obtain the high-probability bound theorem. We will add the detailed theorem and proof in the final version.
>
> **W3. Explanations for the scale**$K$
> Recall that in Theorem 4.5 (page 5), for any $K<\Big\lfloor\min\Big\\{\frac{\rho}{r+1}\frac{\overline{|A_{ij}^0|}}{\eta_0 B},\frac{A_{ii}^0}{\eta_0 B}{\rm for\ }i=1,\cdots,d\Big\\}\Big\rfloor$, Markov heads preserve. In remark 4.8, during the observations in experiments, we found that $\eta_0 = 1e-4$, $B = 1e-6$, $\overline{|A^0_{ij}|} = 1e-2$ and we set $r=20$ then $\rho = 3.61$. Since $A^0_{ii}$ are larger than $\overline{|A^0_{ij}|}$, we can ignore the second part in this inequality. By all the parameters, we can conclude that for any $K < 1.7 e7$, theorem 4.5 will be satisfied. In our setting, we let $K = 1e4$ which is within the theoretical range.
>
> **Q1. In long-term task, why GPT-DTMA cannot surpass DT?**
> In our theoretical results, we demonstrate that Markov heads tend to disproportionately attend to the final input token, thereby impairing long-term planning capabilities. Although the incorporation of a gating mechanism can attenuate the influence of these heads, it does not eliminate them entirely. Another contributing factor may be the difference in training corpora: DT is trained exclusively on a reinforcement learning corpus, whereas GPT-DTMA fine-tunes weights pre-trained on a language modeling corpus. Furthermore, the number of trainable parameters plays a nontrivial role in model performance. Consequently, DT tends to exhibit superior empirical performance. The primary motivation for proposing GPT-DTMA is to enhance the long-term planning ability of GPT-based decision transformers while maintaining lower computational costs by circumventing training from scratch.
>
> **Q2. Would the positional embedding play a role in yielding Markov heads?**
> We acknowledge that positional embeddings may also contribute to the model’s tendency to attend more to the final input token, however, positional embeddings are not directly related with Markov heads. It is important to note that neither DT nor GPT-DT employs positional embeddings. Our analysis is confined to the matrix product $W_q W_k^T$, which is independent of any positional encoding. We define $W_q W_k^T$ as the Markov matrix. In Theorem 4.3 (page 4), we demonstrate that for any random embedding matrix $E$, the transformed matrix $E W_q W_k^T E^T$ remains a Markov head in expectation.
>
> Thanks again for your constructive suggestions and we will include them in the final version.  We are looking forward to further discussions.

---

> > ### Comment · Reviewer_L8Sr · 2025-04-02
> >
> > Thank you for the supplementary experiments and the intuition on theorem. I appreciate them and will increase my rating.
> >
> > Additionally, I think both DT and GPT-DT employ positional embeddings. Please see https://github.com/kzl/decision-transformer/blob/master/gym/decision_transformer/models/decision_transformer.py line 40

---

> > > ### Author Response · Authors · 2025-04-04
> > >
> > > Thank you so much for your acknowledgment and for increasing the rating — we truly appreciate it! Regarding the position embedding, we would like to express more precisely: DT and GPT-DT do not use the original positional embeddings from GPT directly. Instead, they utilize timestep embeddings that serve as their form of positional encoding. These embeddings are also independent of Markov heads, which is consistent with our discussion in Q2. Thank you again for your thoughtful clarification!

---

### Official Review · Reviewer_cgss · 2025-03-09

**Overall Recommendation:** 4

**Summary:**

The paper identifies Markov heads in Pretrained Language Models (PLMs), attention heads with extreme focus on the most recent token. These heads transfer to Decision Transformers (DTs) in offline Reinforcement Learning (RL), improving short-term (Markovian) tasks but harming long-term planning. The paper introduces GPT-DTMA, which uses a Mixture of Attention (MoA) mechanism to adaptively weigh heads, balancing short-term and long-term performance. Theoretical analysis proves Markov heads persist under fine-tuning and experiments confirm GPT-DTMA improves performance across various RL tasks.

**Claims And Evidence:**

I find the claims supported by convincing evidence.

1. **Markov heads exist in PLMs and aid short-term RL.**
- The importance score shows GPT-DT's key heads have diagonal-dominant weight matrices.
- GPT-DT outperforms standard DT in short-horizon MuJoCo tasks.
- CLIP-DT lacks Markov heads and performs worse, confirming their significance.

2. **Markov heads hurt long-term planning**
- GPT-DT underperforms on PointMaze/AntMaze (long-horizon), taking more steps.
- GPT-DTMA's MoA down-weights Markov heads, improving performance in long tasks.

3. Markov heads persist through fine-tuning
- Theorems show that Markov matrices remain dominant after small-gradient updates
- Empirical results confirm that attention remains last-token-focused post finetuning.

**Essential References Not Discussed:**

This paper discusses most related references. One I may suggest is

1. **Trajectory Transformer**
Janner et al., Offline Reinforcement Learning as One Big Sequence Modeling Problem, 2021, for long-horizon transformer planning

**Experimental Designs Or Analyses:**

The experimental design and analyses seem valid. The only thing I would suggest is to make comparisons to non DT-based offline RL methods (e.g., Decision Convformer or value-based methods)

### Datasets
- MuJoCo (short-term)
- Point/Ant Maze (long-term)

### Key findings
- GPT-DTMA matches GPT-DT in short-horizon tasks while significantly improving long-horizon tasks.
- GPT-DTMA-R degrades in short-term, confirming Markov heads aid short-horizon.
- CLIP-DT performs worse than GPT-DT, proving Markov heads are beneficial.

### Baselines
- I like that the authors choice of baselines to control pertraining (GPT-DT), architecture (DTMA), and alternative PLMs (CLIP-DT)
- I suggest the authors to make a comparison to Decision Convformer (Kim et al., 2024), since the paper also discusses Markov properties of short-horizon offline RL tasks.
- It would be great if the authors could make comparisons to value-based offline RL methods (e.g., CQL, IQL) since they outperform Transformer based methods on sub-optimal datasets.

**Methods And Evaluation Criteria:**

I find the methods and evaluation criteria make sense.

### Methods
- Defines Markov heads as diagonal-dominant attention heads.
- Identifies Markov heads by measuring zeroed-out head importance.
- GPT-DTMA introduces gated MoA, letting the model adapt head weights dynamically.

### Evaluation
- Short-horizon tasks: Standard offline MuJoCo locomotion tasks (normalized return)
- Long-horizon tasks: Standard offline Ant-Maze, Point-Maze tasks (steps to goal)
- Baselines: DT, GPT-DT, DTMA, GPT-DTMA, CLIP-DT, GPT-DTMA-R (penalizing Markov heads)
- Robustness: mean and standard deviation over multiple seeds

**Other Comments Or Suggestions:**

I enjoyed this work, and some minor suggestions that can improve this work, or some follow-up work ideas I can think of are as follows.

1. Test other PLMs to confirm generality.

2. Analyze MoA behavior per time step (how weights change dynamically).

3. Compare GPT-DTMA’s performance against value-based offline RL baselines (CQL/IQL).

**Other Strengths And Weaknesses:**

### Strengths

- Original: First work identifying Markov heads and their role in RL.

- Significant: Improves transformer-based RL adaptivity across different timescales.

- Clarity: Good structure; clear motivation and results.

- Balanced theory & practice: Strong theoretical foundation + empirical validation.

### Weaknesses

- Limited PLM scope: Only tested on GPT-2 small; unclear if Markov heads exist in other PLMs.

- Complexity: MoA adds parameters, though minimal overhead.

- No direct comparison to RL SOTA: Does not benchmark against CQL/IQL in D4RL.

- Minor clarity issues: Typos and vague phrasing (e.g., "unknown-term environments").

**Questions For Authors:**

Here are some questions that I found confusing.

1. How was the threshold for Markov heads chosen?

2. Would a manual reduction of Markov head influence improve long-term tasks further?

3. Have you tested Markov heads in other PLMs?

**Relation To Broader Scientific Literature:**

This work is related to broader scientific literature and contributes as follows.

- Background: Decision Transformers, PLM-based RL
- Novel insight: identifies Markov heads as the mechanism behind PLM gains in RL.
- Unique approach: MoA provides adaptive attention, unlike past methods relying on trajectory planning (Trajectory Transformer) or hierarchical policies.
- Contribution to NLP: connects transformer interpretability (head specialization) with RL decision-making.

**Theoretical Claims:**

This paper is not a fully empirical paper. To support their main claim about Markov heads, this paper provides two theorems, which I found reasonable with correct proofs.

1. Theorem 4.3: Markov heads retain their property under random projection (embedding changes).

2. Theorem 4.5: Markov heads remain stable under bounded-gradient finetuning.

---

> ### Author Rebuttal · Authors · 2025-04-01
>
> Thank you for your comprehensive review and constructive comments. We appreciate your acknowledgement regarding the originality, significance, clarity and presentation of our work. Please find our response to suggestions and questions.
>
> **S1&Q3. Test other PLMs to confirm generality.**
> We have tested the existence of Markov heads in other pre-trained large models, such as GPT-J, ImageGPT [1]. We examined the attention heads in GPT-J and found that all of them cannot satisfy the condition (i) in Definition 4.1, i.e. Markov head is not detected. We also test the performance of initializing DT with GPT-J and ImageGPT checkpoints for short-term environments. Table R1 shows the result comparisons among different PLMs. Results show that without Markov heads, the performances of GPTJ-DT and ImageGPT-DT fail to align with GPT-DT.
>
> **Table R1**:
> |Dataset(short-term)|GPT-DT|GPTJ-DT|ImageGPT-DT|
> |-|-|-|-|
> |Hopper-m|77.9|72.5|7.3|
> |Hopper-m-r|77.9|73.8|7.6|
> |Walker2d-m|77.1|75|1.2|
> |Walker2d-m-r|74.0|70.3|12.3|
>
> [1] Chen, Mark, et al. "Generative pretraining from pixels." *International conference on machine learning*. PMLR, 2020.
>
> **S2. Analyze MoA behavior per time step (how weights change dynamically).**
> We analyzed the step-wise changes of average weights for Markov heads and non-Markov heads. Fig.11 and Fig.12 in [[URL](https://anonymous.4open.science/r/submission9905-7D09/readme.md)] show the results for short-term and long-term environment. We conclude that, for short-term environment, the weights of Markov heads gradually ascent while the weights of non-Markov heads gradually descent. In contrast, for long-term environment, the weights of Markov heads gradually descent while the weights of non-Markov heads ascent.
>
> **S3. Compare GPT-DTMA’s performance against value-based offline RL baselines.**
> We have conducted experiments on value-based offline RL methods (CQL) and Decision Convformer (DC). Table R2 shows results for short-term environments, in which CQL generally perform worse than GPT-DT or GPT-DTMA, and DC outperforms GPT-DTMA. Table R3 shows results for long-term environments, where CQL outperforms other methods and DC performs worst. While the results of DC imply that directly utilizing the Markov property benefits short-term environment performances and hurts long-term performances, they strengthened our claim.
>
> **Table R2**:
> |Dataset(short-term)|CQL|DT|GPT-DT|DC|GPT-DTMA|
> |-|-|-|-|-|-|
> |Hopper-m|58.1|67.4|77.9|**79.5**|77|
> |Hopper-m-r|75.3|74.1|77.9|**82.1**|80.4|
> |Walker2d-m|72.7|74.3|77.1|79.3|**79.9**|
> |Walker2d-m-r|78.6|71.9|74.0|**79.1**|77.0|
>
> **Table R3**:
> | Dataset (long-term) | CQL | DT | GPT-DT | DC | GPT-DTMA |
> | --- | --- | --- | --- | --- | --- |
> | PointMaze-large | **167.7** | 195.3 | 257.3 | 276 | 203.0 |
>
> **S4. Extra related works**
> Thanks for pointing out. Decision Convformer (DC) explicitly trains convolutions to enhance local attention; Trajectory Transformer employs a different trajectory formation than DT. In contrast with these work, we identify Markov heads in PLM that influences the short-term and long-term planning ability. We will include these discussions in our final version.
>
> **Q1. How was the threshold for Markov heads chosen?**
> We begin by recalling the definition of a Markov matrix as stated in Definition 4.1: all diagonal elements are positive, and the ratio $\frac{\overline{|A_{ii}|}}{\overline{|A_{ij}|}} > r$. Let $\overline{|A_{ii}|} = m$ and $\overline{|A_{ij}|} = n$, then after applying softmax function, the diagonal element becomes $\frac{e^m}{e^m + (d-1) e^n}$, where $d$ is the embedding dimension. To determine a reasonable range for $r$, we assume that the diagonal element $\frac{e^m}{e^m + (d-1) e^n}$ should be at least $0.5$ in order to express higher attention on last input token, i.e., $\frac{e^m}{e^m + (d-1) e^n} > 0.5$. Then we can obtain that $r > \ln(d-1) + 1$. In our setting, $d = 768$ and $r > 7.64$. In remark 4.7, we set $r=20$ to identify Markov heads of GPT-DT. Under this threshold, the corresponding diagonal element is at least 0.99, indicating an extreme focus on last input token.
>
> **Q2. Would a manual reduction of Markov head influence improve long-term tasks further?**
> A manual reduction of Markov head influence is shown to improve performance on long-term tasks. We investigated this approach using Attention with Reverse Linear Biases (ALiBi-R). ALiBi-R introduces a manual reduction of Markov head behavior by penalizing diagonal elements and simultaneously enhancing the weights assigned to distant elements in the attention matrix.
>
> **Table R4**:
> | Dataset (long-term) | DT | GPT-DT | GPT-DTMA | GPT-DT-ALiBi-R |
> | --- | --- | --- | --- | --- |
> | PointMaze-large | 195.3 | 257.3 | 203.0 | 216 |
>
> For the minor clarity issues, we will revise and address this issues accordingly.
>
> Thank you again for your constructive comments!

---

### Official Review · Reviewer_vRU6 · 2025-03-15

**Overall Recommendation:** 3

**Summary:**

Incorporating Pretrained Language Models (PLMs) into Decision Transformers (DTs) has shown promise in the area of offline reinforcement learning (RL). However, it is unclear why the representations obtained from NLP tasks would be beneficial for RL tasks. The authors aim to address this question by analyzing the attention heads of several models and demonstrating that they exhibit Markov properties. Identifying whether these heads are Markov or not is crucial for understanding the limitations of these models in solving short-term and long-term tasks in various environments.

The authors perform several experiments in both short- and long-term environments and show the impact of Markov heads on model performance. They also conduct various ablation studies to further support their findings, such as examining the relationship between context length and Markov head weight, and initially down-weighting Markov heads to see how this directly affects performance.

**Claims And Evidence:**

The authors claim that PLMs influence performance in offline RL based on Markov heads, which are the key information transferred from PLMs. They argue that this is beneficial only for short-term environments and has a negative impact on long-term environments. The evidence that the authors show is based on empirical observation that supports their theory.

The evidence is not entirely convincing; there has been a lot of research on transformer attention regarding the significance of each head and the observation that not all attention heads are necessary. However, this paper seems disconnected from that body of work. Additionally, it is unclear whether the issue the authors are addressing is related to a distribution shift or a modeling problem, as the paper does not clarify how the short- and long-term data splits were determined. It also does not specify whether the model was trained and tested in the same or different environments.

**Essential References Not Discussed:**

- MOH: MULTI-HEAD ATTENTION AS MIXTURE-OF-HEAD ATTENTION by Jin et al. 2024
- Are Sixteen Heads Really Better than One?  by Michel et al. 2019

**Experimental Designs Or Analyses:**

The author conducted several experiments and analyses to support the claims in their paper:

- Main Results: The main experiments were performed on both short-term tasks (using Mujoco) and long-term tasks (using PointMaze and AntMaze). The authors compared their results to several baselines, which were trained using PLM and from scratch. They analyzed their findings and briefly explained that their proposed algorithm outperforms the others.

- Ablation Studies: The primary focus of the authors’ ablation studies was to compare the relationship between context length and Markov head weights in both short-term and long-term tasks. Although they demonstrated that Markov head weights are higher for short-term than long-term tasks, the paper does not include sample ablations for other models, making it difficult to fully understand the differences.

**Methods And Evaluation Criteria:**

The proposed method and evaluation criteria make sense for the problem at hand.

**Other Comments Or Suggestions:**

N/A

**Other Strengths And Weaknesses:**

N/A

**Questions For Authors:**

1. Would a MoE (mixture of experts) perform better in these long-term environments?

2. What is the high-level intuition behind a Markov head? Additionally, why is diagonal dominance important in the context of NLP and RL? I'm a bit confused about the significance of having an attention head that exhibits diagonal dominance. I would assume some attention heads display diagonal dominance while others do not, as each attention head focuses on different patterns within the data.

3. How did you use for the train and test splits when training DT and GPT-DT? Did you train and test on different datasets from the same environment, or did you train in one environment and test in another?

4. Not all attention heads are Markov heads. How do the non-Markov heads affect performance in both long-term and short-term environments? In general, what percentage of heads are considered Markov heads? Are Markov heads fixed within a particular environment, or do the heads classified as Markov change depending on the environment?

5. Section 4.2: "In real applications, it is hard to determine whether the planning ability is required by the current environment without prior knowledge". I am uncertain whether the issues discussed stem from distribution shift problems due to changes in the training and testing environments or if they are related to modeling flaws.

6. Why do you need to adaptively control the weight of the attention heads? It does not seem like you are addressing the modeling problem.

7. Do you have a skyline model that shows what good behavior looks like, even if it's achieved through cheating? For example, imagine training a model on certain tasks and then testing it on different tasks. A skyline model would be trained on the test tasks to observe the behavior of the model as if it had seen the training datasets.

8. How does the Markov heads change for DT and GPT-DT similar to the experiments shown in table 5 and table 6?

9. I assume that with an increase in context length, regardless of the model, the models will tend to focus more easily on distant information because there is more content to consider. This seems like a fairly general phenomenon that is not specific to GPT-DTMA.

**Relation To Broader Scientific Literature:**

The key contribution to the broader scientific literature is understanding when the representations of PLMs will be helpful in solving RL tasks. In particular, the papers suggest that PLMs learn what are referred to as Markov Heads, which RL tasks can take advantage of. However, these Markov Heads may only be adequate for short-term tasks and not for long-term tasks. Therefore, the broader scientific connection is to determine when PLMs will be beneficial for solving RL tasks.

**Theoretical Claims:**

No, I did not check the correctness of the proofs in the paper.

---

> ### Author Rebuttal · Authors · 2025-04-01
>
> Thanks for your valuable comments. Please kindly find the response to your concerns below.
>
> **Q1. Would a MoE (mixture of experts) perform better in these long-term environments?**
> It’s interesting to investigate whether MoE perform well in long-term environments, however, MoE is not considered in this work due to the main focus of this work is on unveiling the impact of Markov heads in PLMs that affect offline RL performances. Indeed, Markov heads may exist in each expert of an MoE model, so our findings may also provide insights for explaining the performance of MoE in long-term environments.
>
> **Q2. What is the high-level intuition behind a Markov head? Additionally, why is diagonal dominance important in the context of NLP and RL?**
> The underlying intuition behind Markov heads is that they function as strong local policy learners, relying predominantly on the most recent observation to inform decisions. This behavior mirrors the memoryless property of Markov processes. Diagonal dominance is a critical characteristic, as it ensures that attention is primarily focused on the current input token, thereby reinforcing the Markovian assumption and promoting stable learning in highly reactive environments. While we acknowledge that not all attention heads exhibit diagonal dominance, we demonstrate that this variation reflects the emergence of diverse temporal planning behaviors. Specifically, only a subset of heads learns to follow short-term decision patterns, a phenomenon particularly pertinent in reinforcement learning settings where tasks may exhibit heterogeneous temporal dependencies.
>
> **Q3. How did you use for the train and test splits when training DT and GPT-DT? Did you train and test on different datasets from the same environment, or did you train in one environment and test in another?**
> Following common practice in offline RL research, we train each model with offline datasets collected by D4RL for each environment respectively, and test the model in the same environment.
>
> **Q4. How do the non-Markov heads affect performance in both long-term and short-term environments? In general, what percentage of heads are considered Markov heads? Are Markov heads fixed within a particular environment, or do the heads classified as Markov change depending on the environment?**
> Non-Markov heads show almost equal attention on each tokens, and we can see that the weights of all non-Markov heads become larger in long-term environments, such as PointMaze tasks. During our observations, we found that 3 of 12 heads are Markov heads. The Markov Heads are defined according to Definition 4.1 and 4.2 regardless of environments. According to Theorem 4.5, for GPT-DTs, this Markov Head property are fixed and won’t change by fine-tuning. For DT, the model is trained to obtain or not obtain Markov Heads based whether it’s short-term or long-term environment.
>
> **Q5. Is the issue discussed in Section 4.2 stem from distribution shift problems due to changes in the training and testing environments or related to modeling flaws.**
> We would like to clarify that the issues discussed are neither a distribution shift problem nor related to modeling flaws. We state that when given an new task for a PLM initialized DT model to be fine-tuned on, we may not know the planning ability required (either short-term or long-term) for this task in advance. If long-term planning ability is required, we have proved that the Markov Heads in PLM weights will hurt performance.
>
> **Q6. Why do you need to adaptively control the weight of the attention heads?**
> Due to the issue stated in Section 4.2 and given our statement in Q5, we would like the model to automatically adapt to the planning ability needed. Therefore, GPT-DTMA learns a weight reducing the influence of Markov heads to enhance long-term planning ability of GPT-DTs.
>
> **Q7. Do you have a skyline model that shows what good behavior looks like?**
> While our experiments follow the standard offline RL settings, trainings and testings are conducted on the same task for each model. Therefore, the current results have represented the “good behavior” given each model configuration.
>
> **Q8. How does the Markov heads change for DT and GPT-DT as in Table 5/6?**
> We would like to clarify that Markov head is not observed in DT. Table 5/6 shows the head weights given by the MoA module in GPT-DTMA, therefore they are not available for DT and GPT-DT.
>
> **Q9. With an increase in context length would the models tend to focus more easily on distant information?**
> In Theorem 4.3 and 4.5, we prove that Markov heads persists in GPT-DT during fine-tuning, so it will not tend to focus on distant information natively, even when context length is increased. Therefore, the performance gain brought by MoA to GPT-DTMA is non-trivial and cannot be replaced by simply extending the context length.
>
> Thank you again for your reviewing efforts and we sincerely hope our responses have addressed your concerns.

---

### Official Review · Reviewer_WJgz · 2025-03-27

**Overall Recommendation:** 3

**Summary:**

This paper investigates why pre-trained language models (PLMs) boost Decision Transformer performance in offline RL setting. The authors identify crucial "Markov heads" within PLMs that strongly focus attention on the most recent input state. While beneficial for short-term tasks like MuJoCo, theoretical analysis and experiments show these heads are rigid and cannot be easily changed via fine-tuning, thus hindering performance in long-term planning tasks like Mazes.

To address this limitation, the paper introduces GPT-DTMA, which uses a Mixture of Attention mechanism to adaptively weight the influence of different attention heads. This allows the model to dynamically control the impact of Markov heads based on the specific environment's requirements. Results demonstrate that GPT-DTMA outperforms baselines in short-term settings and significantly closes the performance gap in long-term scenarios.

**Claims And Evidence:**

The paper does a good job supporting the claims.

For experimental evidence, the main issue is the improvement of GPT-DTMA—for most cases in Table 3, the improvement is within the confidence interval, which weakens the validity of the proposed algorithm. Even for long-term scenario in table 4, the case is not that better.

**Essential References Not Discussed:**

All references are discussed.

**Experimental Designs Or Analyses:**

I have checked the validity of experiments, and most are reasonable, clear. My only issue is the significance of table 3/4.

**Methods And Evaluation Criteria:**

The proposed methods overall make sense. For the benchmark environment, do you have any other experiments beyond the robot environment or maze environment? They are relatively old environments in the offline RL setting.

**Other Comments Or Suggestions:**

No

**Other Strengths And Weaknesses:**

Strengths:
The overall idea is interesting and the demonstration process is quite clear and sound. The proposed Markov Head Concept is novel and properly addressed.

weakness:
While GPT-DTMA shows improvements over GPT-DT and DT baselines, the paper doesn't extensively compare its absolute performance against the broader state-of-the-art in offline RL methods on these benchmarks (which might include non-Transformer methods or different Transformer variants). The focus is more internal to understanding PLM transfer within the DT framework.

**Questions For Authors:**

1. Your analysis focuses primarily on GPT-2. Have you conducted preliminary investigations or do you have hypotheses about whether similar Markov head phenomena exist and exhibit the same stability when using other PLM architectures?
2.  The Markov heads are based on a ratio r (in Definition 4.1/4.2 context). How sensitive are your findings – specifically, the number of heads identified as Markov and the overall conclusions about their impact – to the choice of this threshold r?

**Relation To Broader Scientific Literature:**

It is closely related with the LLM and pretraining literature. The author properly discuss its relationship in the related work and preliminary section.

**Theoretical Claims:**

I didn't check the details but the overall logics seem fine. My main point is that the author should address more on how some of the assumptions/conditions are determined. For example, for more explanation on the physical meaning of definition 4.1 and 4.2 can make it more reader-friendly.

---

> ### Author Rebuttal · Authors · 2025-04-01
>
> Thanks for your valuable comments. Please kindly find the response to your concerns below.
>
> **W1. The improvement of GPT-DTMA is within the confidence interval.**
> The experiment are repeated three times to ensure significance. We would also like to emphasize that, one important objective for our experiments is to validate the influence of Markov heads in short-term and long-term environments. Since Markov heads play important role in both GPT-DT and GPT-DTMA, it is reasonable that their performance stay close in short-term environments. However, their performance gap in long-term environment is larger, showing support to our main claim.
>
> **W2. The paper doesn't extensively compare its absolute performance against the broader state-of-the-art in offline RL methods on these benchmarks.**
> We have extended experiments to compare with value-based offline RL methods (CQL) and Decision Convformer in our results. Please refer to Table R2 and Table R3. Table R2 shows results for short-term environments, in which CQL generally perform worse than GPT-DT and GPT-DTMA and DC outperforms GPT-DTMA. Table R3 shows results for long-term environments, where CQL outperforms other methods and DC performs worst. While the results of DC imply that directly utilizing the Markov property benefits short-term environment performances and hurts long-term performances, they strengthened our claim.
>
> **Table R2**:
> | Dataset (short-term) | CQL | DT | GPT-DT | DC | GPT-DTMA |
> | --- | --- | --- | --- | --- | --- |
> | Hopper-m | 58.1 | 67.4 | 77.9 | **79.5** | 77 |
> | Hopper-m-r | 75.3 | 74.1 | 77.9 | **82.1** | 80.4 |
> | Walker2d-m | 72.7 | 74.3 | 77.1 | 79.3 | **79.9** |
> | Walker2d-m-r | 78.6 | 71.9 | 74.0 | **79.1** | 77.0 |
>
> **Table R3**:
> | Dataset (long-term) | CQL | DT | GPT-DT | DC | GPT-DTMA |
> | --- | --- | --- | --- | --- | --- |
> | PointMaze-large | **167.7** | 195.3 | 257.3 | 276 | 203.0 |
>
> **Q1. Your analysis focuses primarily on GPT-2. Have you conducted preliminary investigations or do you have hypotheses about whether similar Markov head phenomena exist and exhibit the same stability when using other PLM architectures?**
> We have tested the existence of Markov heads in other pre-trained large models, such as GPT-J, ImageGPT [1]. We examined the attention heads in GPT-J and found that all of them cannot satisfy the condition (i) in Definition 4.1, i.e. Markov head is not detected. We also test the performance of initializing DT with GPT-J and ImageGPT checkpoints for short-term environments. Table R1 shows the result comparisons among different PLMs. Results show that without Markov heads, the performances of GPTJ-DT and ImageGPT-DT fail to align with GPT-DT.
>
> [1] Chen, Mark, et al. "Generative pretraining from pixels." International conference on machine learning. PMLR, 2020.
>
> **Table R1**:
> | Dataset (short-term) | GPT-DT | GPTJ-DT | ImageGPT-DT |
> | --- | --- | --- | --- |
> | Hopper-m | 77.9 | 72.5 | 7.3 |
> | Hopper-m-r | 77.9 | 73.8 | 7.6 |
> | Walker2d-m | 77.1 | 75 | 1.2 |
> | Walker2d-m-r | 74.0 | 70.3 | 12.3 |
>
> **Q2. The Markov heads are based on a ratio r (in Definition 4.1/4.2 context). How sensitive are your findings – specifically, the number of heads identified as Markov and the overall conclusions about their impact – to the choice of this threshold r?**
> We begin by recalling the definition of a Markov matrix as stated in Definition 4.1: all diagonal elements are positive, and the ratio $\frac{\overline{|A_{ii}|}}{\overline{|A_{ij}|}} > r$. Let $\overline{|A_{ii}|} = m$ and $\overline{|A_{ij}|} = n$, then after applying softmax function, the diagonal element becomes $\frac{e^m}{e^m + (d-1) e^n}$, where $d$ is the embedding dimension. To determine a reasonable range for $r$, we assume that the diagonal element $\frac{e^m}{e^m + (d-1) e^n}$ should be at least $0.5$ in order to express higher attention on last input token, i.e., $\frac{e^m}{e^m + (d-1) e^n} > 0.5$. Then we can obtain that $r > \ln(d-1) + 1$. In our setting, $d = 768$ and $r > 7.64$. In remark 4.7, we set $r=20$ to identify Markov heads of GPT-DT. Under this threshold, the corresponding diagonal element is at least 0.99, indicating an extreme focus on last input token. From Table 1, we can see that there are three Markov heads before/after fine-tuning for any $r\in(20,+\infty)$.
>
> Thank again for your thoughtful reviews. We will incorporate the discussions in the revised version and we are looking forward to further discussions.

---

### Decision · Program_Chairs · 2025-05-01

**Decision:**

Accept (poster)

**Comment:**

This paper identifies that Markov heads in a PLM can transfer to a DT in offline RL, improving performances on short-term environments while degrading performances on long-term environments. Based on this observation, it proposes the mixture of attention (MoA) mechanism to adaptively weight different attention heads so that it especially down-weights Markov heads for long-term planning tasks. The empirical and theoretical analysis on the relationship between the existence of Markov heads and RL performances of PLM-based DT would be interesting, and the proposed MoA would be technically sound with sufficient empirical validation. Most concerns raised by the reviewers are addressed by the authors. Based on the consensus among the reviewers, I would recommend this paper to be accepted.

I think more extensive empirical validation with recent small language models and on diverse real RL tasks would be helpful for this paper to be more valuable. For example, the performance gaps for short-term environments between GPT-DT and GPTJ-DT seems to be small even though there is no Markov head in GPTJ-DT.